# EFFICIENT PRECISION AND RECALL METRICS FOR ASSESSING GENERATIVE MODELS USING HUBNESS-AWARE SAMPLING

## ABSTRACT

Despite impressive results, deep generative models require massive datasets for training. As dataset size increases, effective evaluation metrics like precision and recall (P&R) become computationally infeasible on commodity hardware. In this paper, we address this challenge by proposing efficient P&R (eP&R) metrics that give almost identical results as the original P&R but with much lower computational costs. Specifically, we identify two redundancies in the original P&R: i) redundancy in ratio computation and ii) redundancy in manifold inside/outside identification. We find both can be effectively removed via hubness-aware sampling, which extracts representative elements from synthetic/real image samples based on their hubness values, *i.e.*, the number of times a sample becomes a $k$-nearest neighbor to others in the feature space. Thanks to the insensitivity of hubness-aware sampling to exact $k$-nearest neighbor ($k$-NN) results, we further improve the efficiency of our eP&R metrics by using approximate $k$-NN methods. Extensive experiments show that our eP&R matches the original P&R but is far more efficient in time and space.

## 1 INTRODUCTION

Deep generative models have achieved great success by combining deep learning with generative modeling. However, they have also inherited the data-hungry nature of deep learning, requiring massive datasets for training. For instance, the FFHQ dataset used to train StyleGAN contains 70 thousand images (Karras et al., 2019), while the Latent Diffusion model leveraged LAION-400M's 400 million text-image pairs (Rombach et al., 2022a). Stable Diffusion[1] pushed this even further, training its models on LAION-5B's 5 billion pairs (Schuhmann et al., 2022). Despite their impressive results, the massive scale of datasets used to train modern deep generative models presents challenges for evaluation. As dataset size increases, some of the most effective evaluation metrics (Salimans et al., 2016; Heusel et al., 2017; Sajjadi et al., 2018; Simon et al., 2019; Kynkäänniemi et al., 2019), which compare generated and real image distributions, may become computationally infeasible for commodity GPUs and ordinary research institutions. Going forward, developing more efficient evaluation metrics becomes critical to continue advancing the state of the art (SOTA).

Among the most effective evaluation metrics, Fréchet Inception Distance (FID) (Heusel et al., 2017) and Inception Score (IS) (Salimans et al., 2016) are relatively computationally efficient. Let $n$ be the number of samples, they have linear time and space complexity of $O(n)$, as they rely on simple statistics of extracted features. Specifically, the feature extraction takes $O(n)$ time and space while the statistics (*e.g.*, mean) computation also takes linear time $O(n)$ for a given feature extractor like Inception v3 (Szegedy et al., 2015) with fixed feature dimensions. However, both FID and IS are one-dimensional scores that cannot distinguish between different failure modes. Addressing this issue, the precision and recall (P&R) metrics (Sajjadi et al., 2018; Simon et al., 2019; Kynkäänniemi et al., 2019) were employed. Intuitively, precision measures the *quality* of synthesized images while recall measures their *diversity*. Although effective, the SOTA version of P&R (Kynkäänniemi et al., 2019) requires costly pairwise distance calculations (*e.g.*, in $k$-nearest neighbor algorithm) between

---

[1] https://stability.ai/stable-diffusion

extracted features of samples and sorting, consuming $O(n^2 \log n)$ time and space, and thus becomes computationally infeasible when evaluating deep generative models trained on large-scale datasets.

In this work, we address the high computational costs of precision and recall (P&R) metrics with a novel solution based on *hubness-aware sampling*. Specifically, we have identified two important types of *redundancies* in the computation of P&R: i) redundancy in the P&R ratio computation and ii) redundancy in identifying whether a sample is within or outside of a manifold (*e.g.*, synthetic or real image manifold). Interestingly, we find that both these redundancies can be effectively removed by hubness-aware sampling. In a nutshell, hubness-aware sampling extracts a small number of $m$ representative elements from the real/synthetic samples based on their *hubness values*, defined as the number of times a sample becomes a $k$-nearest neighbor ($k$-NN) to others in the feature space (Radovanovic et al., 2010; Liang et al., 2022). We denote such representative elements as "hubs", which have higher hubness values than their peers. We conjecture that the validity of our approach comes from the fact that hubness values are effective importance identifiers for samples with respect to the $k$-NN results on which P&R (Kynkäänniemi et al., 2019) relies. In addition, utilizing the fact that the identification of hubness points relies on their relatively higher hubness values rather than exact $k$-NN results, we further improve the efficiency of our eP&R metrics using approximate $k$-NN methods. Extensive experimental results demonstrate that the P&R calculated using such representative elements is almost identical to the original P&R, but consumes much less time and space. Our contributions include:

- We propose the *efficient precision and recall* (eP&R) metrics for assessing generative models, which give almost identical results as the original P&R metrics (Kynkäänniemi et al., 2019) but consume much less time and space. Theoretically, our eP&R metrics run in $O(mn \log n)$ time and consume $O(mn)$ space ($m$ is the number of of hubs samples and $m \ll n$), which are much more efficient than the original P&R metrics that run in $O(n^2 \log n)$ time and consumes $O(n^2)$ space.

- We identify two important types of redundancies in the original P&R metrics and uncover that both of them can be effectively removed by hubness-aware sampling (Radovanovic et al., 2010; Liang et al., 2022). In addition, the insensitivity of hubness-aware sampling to exact $k$-nearest neighbor ($k$-NN) results allows for further efficiency improvement by using approximate $k$-NN methods.

- Extensive experimental results demonstrate the effectiveness of our eP&R metrics.

## 2 RELATED WORK

### 2.1 DEEP GENERATIVE MODELS

Deep generative models have achieved impressive results in image-generation tasks. Major models include Variational Autoencoders (VAEs), Generative Adversarial Networks (GANs), and diffusion models, which will be reviewed below respectively.

**Variational Autoencoders (VAEs).** VAEs use variational inference to approximate posterior inference, training an encoder network to map inputs to a latent space and a decoder network to reconstruct the inputs from the latents (Kingma & Welling, 2013). Despite their elegant theory, images generated by early VAEs are usually blurry, which was improved by incorporating latent quantization to produce models like VQ-VAE (Van Den Oord et al., 2017) and VQ-VAE2 (Razavi et al., 2019) that can synthesize sharp and high-resolution images.

**Generative Adversarial Networks (GANs).** GANs (Goodfellow et al., 2014) train two neural networks concurrently: a generator network that produces synthetic outputs, and a discriminator network that distinguishes real from synthetic data. The two networks are pitted against each other in a minimax adversarial game, where the generator tries to fool the discriminator and the discriminator tries to identify fakes. This creates a constant evolutionary pressure that enables GANs to produce increasingly realistic outputs. A core innovation of GANs is using the discriminator not just for evaluation, but directly in the training loop to guide the generator. GANs can produce sharp and photorealistic images, but are notoriously difficult to train due to mode collapse, optimization instability, and other challenges. This has led to significant efforts to stabilize GAN training (Arjovsky et al., 2017; Gulrajani et al., 2017; Miyato et al., 2018; Zhou et al., 2019; Qin et al., 2020).

Along with these efforts, researchers have extended the synthesis capabilities of GANs to a variety of image generation tasks, including unconditional image synthesis (Karras et al., 2017; 2019; 2020; 2021; Sauer et al., 2022), conditional image synthesis (Mirza & Osindero, 2014; Miyato et al., 2018; Brock et al., 2019), image-to-image translation (Isola et al., 2017; Park et al., 2019; Zhu et al., 2020), image editing (Abdal et al., 2019; 2020), etc.

**Diffusion Models.** Diffusion models (Sohl-Dickstein et al., 2015; Ho et al., 2020; Dhariwal & Nichol, 2021) train a neural network to reverse a stochastic diffusion process. They start with a data sample $x$ and apply a diffusion process that gradually adds Gaussian noise over multiple timesteps to arrive at a noisy sample $\hat{x}$. A neural network is then trained to take $\hat{x}$ and predict the noise that was added at each step, allowing the original $x$ to be reconstructed. By training the model to denoise the diffused samples, it learns to generate high-quality samples. Diffusion models avoid problematic generator-discriminator training and provide exact log-likelihoods. However, sampling requires running the full diffusion process in reverse, which is computationally expensive. Extensions like DDIM (Song et al., 2020) have made diffusion models more efficient. Thanks to their training stability, diffusion models have been widely used in text-to-image synthesis and editing tasks, including the Latent Diffusion model (Rombach et al., 2022a) that inspired Stable Diffusion, DALLE-2 (Ramesh et al., 2022), Imagen (Saharia et al., 2022), DreamBooth (Ruiz et al., 2022), MUSE (Chang et al., 2023).

Nevertheless, despite their success, modern deep generative models require massive datasets for training. This makes their evaluation challenging as the most effective evaluation metrics compare the full distributions of generated and real images, which may become computationally infeasible for commodity GPUs and non-prestigious institutions.

## 2.2 Evaluation Metrics for Assessing Deep Generative Models

**Fréchet Inception Distance (FID).** The FID metric, introduced by Heusel et al. (2017), computes the Fréchet distance between features of the real and generated images extracted by an Inception-V3 feature extractor. A lower FID score indicates a higher similarity between the distributions of real and generated images, implying better image quality and diversity in the generated samples. Thus, the computation of FID consumes $O(n)$ time and space as the feature extraction takes $O(n)$ time and space while the Fréchet distance computation also takes linear time $O(n)$ when using a fixed Inception-V3 network.

**Inception Score (IS).** The IS metric, proposed by Salimans et al. (2016), uses a pre-trained Inception-v3 classification model to compute the conditional label distribution $p(y|x)$ for each generated image $x$. IS measures two main aspects: i) the diversity of generated images, indicated by the entropy of $p(y|x)$, and the precision of generated images, indicated by the KL divergence between the marginal distribution $p(y)$ and the conditional distribution $p(y|x)$ for each $x$. A higher IS generally indicates the model can generate more realistic and diverse images. Similar to that of FID, the computation of IS consumes $O(n)$ time and space as the feature extraction takes $O(n)$ time and space while the computation of IS metric takes linear time $O(n)$.

The FID and IS metrics were improved by Chong & Forsyth (2020) to $\text{FID}_\infty$ and $\text{IS}_\infty$, which apply Quasi-Monte Carlo integration to reduce bias and improve reliability of them for finite samples.

**Precision and Recall (P&R).** Despite their effectiveness, FID and IS metrics are one-dimensional scores and thus cannot differentiate between specific failure modes, *e.g.*, mode dropping or collapsing (Lin et al., 2018), or provide insights into the underlying causes of poor performance. The P&R metrics were employed to address this issue (Sajjadi et al., 2018; Simon et al., 2019; Kynkäänniemi et al., 2019). In short, precision measures the percentage of generated samples that are considered high-quality and indistinguishable from real data, indicating the *quality* of generated samples; recall measures the percentage of all potential high-quality samples that the generator was able to produce, indicating the *diversity* of generated samples. Specifically, Sajjadi et al. (2018) formulated P&R through relative probabilistic densities between the distributions of real and generated images, which are non-trivial to estimate. Addressing this issue, they proposed a practical algorithm based on the maximal achievable values of an alternative definition of P&R. Their method was generalized by Simon et al. (2019) to accommodate arbitrary distributions and link P&R to type I and type II errors of likelihood ratio classifiers. Observing that the P&R implementation proposed by Sajjadi et al. (2018) relies on relative densities and thus cannot correctly identify mode collapse/truncation,

Kynkäänniemi et al. (2019) propose to model the real and generated image manifolds directly using the $k$-nearest neighbors of samples, which is the state-of-the-art (SOTA) version of P&R for assessing generative models. Although more accurate, their method is computationally expensive as $k$-NN consumes $O(n^2)$ time and space, making their metrics infeasible to compute using commodity hardware on the large datasets used by modern deep generative models.

Addressing this issue, we propose the efficient precision and recall (eP&R) metrics based on a novel strategy named *hubness*-aware sampling, which give almost identical results to (Kynkäänniemi et al., 2019) but consume much less time and space.

### 2.3 HUBNESS PHENOMENON

*Hubness* is a widely recognized phenomenon of nearest neighbors search in high-dimensional spaces that arises from the well-known "curse of dimensionality" (Radovanovic et al., 2010). It pertains to the inherent characteristics of data distributions in high-dimensional spaces and reveals a counterintuitive fact: even with uniform distributions, high dimensionality gives rise to the emergence of "popular" nearest neighbors (Newman et al., 1983; Newman & Rinott, 1985; Radovanovic et al., 2010), *i.e.*, points that are significantly more likely to be among the $k$-nearest neighbors of other points within a given sample set, denoted as *hubs*.

To mitigate the effects of the hubness phenomenon, researchers have developed various solutions in different domains, including gene expression classification (Buza, 2016a;b), time-series classification (Tomašev et al., 2015), and electroencephalograph classification (Buza & Koller, 2016). Furthermore, hubness-aware $k$-nearest neighbor ($k$-NN) methods have been developed, including hubness-weighted $k$-NN (Radovanovic et al., 2010), hubness-fuzzy $k$-NN (Tomašev et al., 2014), hubness-information $k$-NN (Tomasev & Mladenic, 2011), Naive Hubness-Bayesian $k$-NN (Tomašev et al., 2011), and Augmented Naive Hubness-Bayesian $k$-NN (Tomašev & Mladenić, 2013). By incorporating measures and adjustments specifically designed for the hubness phenomenon, these methods have improved the accuracy and reliability of $k$-NN algorithms.

However, instead of mitigation, recent works have demonstrated that depending on the task, the hubness phenomenon can be very useful. For example, Liang et al. (2022) showed that the hubness phenomenon can be used as a prior to identify high-quality latents in GAN latent spaces. Following the same philosophy, this work introduces a new method to improve the computational efficiency of P&R metrics by incorporating hubness-aware sampling.

## 3 PRELIMINARIES

As proposed by Kynkäänniemi et al. (2019), the precision and recall (P&R) metrics for assessing generative models are defined as:

$$\text{precision}(\mathbf{\Phi}_r, \mathbf{\Phi}_g) = \frac{1}{|\mathbf{\Phi}_g|} \sum_{\phi_g \in \mathbf{\Phi}_g} f(\phi_g, \mathbf{\Phi}_r), \quad \text{recall}(\mathbf{\Phi}_r, \mathbf{\Phi}_g) = \frac{1}{|\mathbf{\Phi}_r|} \sum_{\phi_r \in \mathbf{\Phi}_r} f(\phi_r, \mathbf{\Phi}_g) \quad (1)$$

where $\mathbf{\Phi_g}$ and $\mathbf{\Phi_r}$ are the sets of feature vectors corresponding to the generated and real image samples, respectively; $|\mathbf{\Phi}|$ denotes the number of samples in set $\mathbf{\Phi}$ and $|\mathbf{\Phi}_g| = |\mathbf{\Phi}_r|$; $f(\phi, \mathbf{\Phi})$ is a binary function determining whether a sample $\phi$ lies on a manifold represented by $\mathbf{\Phi}$ that:

$$f(\phi, \mathbf{\Phi}) = \begin{cases} 1, & \text{if } \|\phi - \phi'\|_2 \leq \|\phi' - \text{NN}_k(\phi', \mathbf{\Phi})\|_2 \text{ for } \textbf{at least one } \phi' \in \mathbf{\Phi} \\ 0, & \text{otherwise}, \end{cases} \quad (2)$$

where $\text{NN}_k(\phi', \mathbf{\Phi})$ denotes the $k$th nearest neighbour of $\phi'$ in $\mathbf{\Phi}$. Intuitively, their precision and recall metrics estimate the generative and real image manifolds with a collection of hyperspheres, respectively, with each feature vector sample as the center and the distance between it and its $k$th nearest neighbor as the radius. A sample $\phi$ is determined to lie on a manifold if it lies within the hyperspheres of that manifold and vice versa.

## 4 EFFICIENT PRECISION AND RECALL

Although effective, Eqs. 1 and 2 are computationally expensive due to the calculation of pairwise distances between samples and the sorting required by $k$-NN, which grows quasi-quadratically with

the number of samples. This prevents them from being computed on large datasets with commodity GPUs and hampers the progress of the field. To improve the computational efficiency of precision and recall (P&R) metrics, we identify two important types of *redundancies* in Eqs. 1 and 2 (Sec. 4.1) and propose to address them using *hubness-aware sampling*, whose insensitivity to exact $k$-NN results allows for further efficiency improvement (Sec. 4.2). We also conduct a computational complexity analysis (Sec. 4.3) to demonstrate the high computational efficiency of our method.

## 4.1 REDUNDANCIES IN PRECISION AND RECALL CALCULATIONS

As mentioned above, we have identified two important types of *redundancies* in P&R calculations: i) redundancy in the P&R ratio computation and ii) redundancy in identifying whether a sample is within or outside of a manifold (*e.g.*, synthetic or real image manifold) as follows:

**Observation 4.1 (Redundancy in Ratio Estimation)** *As Eq. 1 shows, the P&R metrics are essentially ratios of the number of samples in a set $\mathbf{\Phi}$ that lie on a given manifold to the number of all samples in $\mathbf{\Phi}$. Thus, we can obtain similar P&R ratios by using representative samples of $\mathbf{\Phi}$ with the rest as redundant.*

**Observation 4.2 (Redundancy in Inside/Outside Manifold Identification)** *As shown in Eq. 2, $f(\phi, \mathbf{\Phi})$ is 1 as long as $\phi$ is within the $k$-NN hypersphere of at least one sample $\phi' \in \mathbf{\Phi}$. This means that we only need to find one valid $\phi'$ for each $\phi$ and all the other $\phi's$ are redundant.*

## 4.2 REDUNDANCY REDUCTION USING HUBNESS-AWARE SAMPLING

Interestingly, we find hubness-aware sampling to be an effective solution for both redundancies. Specifically, for Observation 4.1, we find that samples with similar hubness values are effective representative samples of set $\mathbf{\Phi}$ in terms of P&R ratios as they share similar ratios of samples identified as 1 *vs.* 0 by Eq. 2 (Fig. 1), indicating that we can use a small number of hubs samples to approximate P&R; for Observation 4.2, we find that most $\phi$ with $f(\phi, \mathbf{\Phi}) = 1$ (Eq. 2) are included in the $k$-NN hypersphere of at least one $\phi'$ with high hubness values, *i.e.*, hubs samples (Fig. 2), indicating that we can obtain similar outputs of Eq. 2 using a small number of hubs samples. Thus, our efficient P&R metrics can be defined as:

$$\text{precision}^{hub}(\mathbf{\Phi}_r, \mathbf{\Phi}_g) = \frac{1}{|\mathbf{\Phi}_g^{hub}|} \sum_{\phi_g^{hub} \in \mathbf{\Phi}_g^{hub}} f(\phi_g^{hub}, \mathbf{\Phi}_r^{hub}) \tag{3}$$

$$\text{recall}^{hub}(\mathbf{\Phi}_r, \mathbf{\Phi}_g) = \frac{1}{|\mathbf{\Phi}_r^{hub}|} \sum_{\phi_r^{hub} \in \mathbf{\Phi}_r^{hub}} f(\phi_r^{hub}, \mathbf{\Phi}_g^{hub}) \tag{4}$$

where $\mathbf{\Phi}_g^{hub}$ and $\mathbf{\Phi}_r^{hub}$ are the sets of feature vectors with hubness values $m > t$ corresponding to the generated and real image samples, respectively; $t$ is a threshold hyper-parameter.

**Efficient Hubs Sample Identification.** Despite their effectiveness, the identification of hub samples is also based on the $O(n^2)$ $k$-NN algorithm which is expensive in both time and space. Fortunately, such identification is insensitive to exact $k$-NN results as it only relies on a rough threshold $t$ of the hubness values. Thus, we can use an approximate $k$-NN algorithm for the identification of hub samples that further improves the efficiency of our metrics.

## 4.3 COMPUTATIONAL COMPLEXITY ANALYSIS

To provide a clear demonstration of the computational efficiency of our metrics, we conduct a computational complexity analysis as follows. Given two sets $\mathbf{\Phi}_r$ and $\mathbf{\Phi}_g$ ($|\mathbf{\Phi}_r| = |\mathbf{\Phi}_g| = n$), the calculation of the original P&R metrics (Kynkäänniemi et al., 2019) can be divided into five stages:

1. **[Distance Matrices of $\mathbf{\Phi}_r$ and $\mathbf{\Phi}_g$]** Calculating pairwise distances for samples in $\mathbf{\Phi}_r$ and $\mathbf{\Phi}_g$ respectively, which consumes $O(n^2)$ time and space for each set.

2. **[Sorting]** Sorting the distance matrices as required by the $k$-NN algorithm, which consumes $O(n^2 \log n)$ time and no extra space.

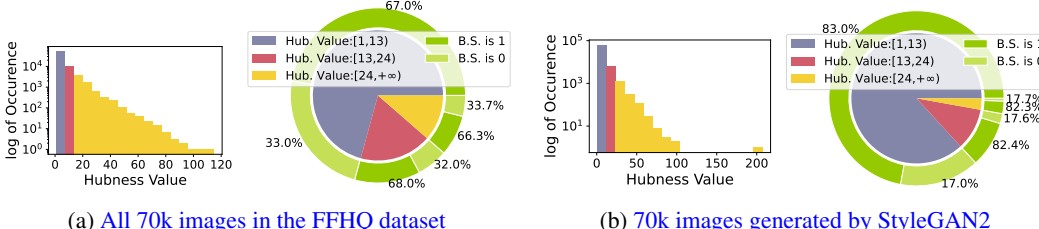

(a) All 70k images in the FFHQ dataset    (b) 70k images generated by StyleGAN2

Figure 1: Samples with similar hubness values are effective representative samples in terms of P&R ratios calculation. (a) Left: Histogram of sample occurrences (log scale) *vs.* hubness value (FFHQ). The samples are grouped into different colors based on similar hubness values. Right: Pie chart showing that all three groups share similar ratios of samples identified as 1 vs. 0 (green *vs.* light green) using Eq. 2 for recall calculation. (b) The same experiment as (a) but on StyleGAN-generated samples for precision calculation. Following (Kynkäänniemi et al., 2019), we use VGG-16 as a feature extractor and StyleGAN2 trained on the FFHQ dataset as the generative model to be assessed. Please see Sec. A.4 for more details. Hub.: Hubness; B.S: binary score.

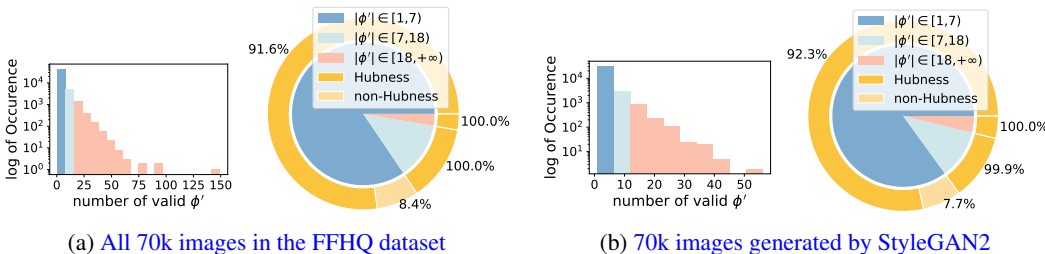

(a) All 70k images in the FFHQ dataset    (b) 70k images generated by StyleGAN2

Figure 2: Most samples $\phi$ with $f(\phi, \Phi) = 1$ (Eq. 2) are included in the $k$-NN hypersphere of at least one hubs sample ($t = 3$) of the other distribution. (a) Left: Histogram of sample occurrences (log scale) *vs.* the times a sample is included in the $k$-NN hypersphere of a sample of the other distribution, *i.e.*, valid $\phi'$ (FFHQ). The samples are grouped into different colors based on similar numbers of valid $\phi'$. Right: Pie chart showing the ratio of samples within the $k$-NN hypersphere of *hubness* vs. *non-hubness* samples from the other distribution, to the total number of samples $\phi$ with $f(\phi, \Phi) = 1$ in each group. Hubness: points with hub values above a threshold $t \geq 3$; Non-hubness: $t < 3$. (b) The same experiment as (a) but on StyleGAN-generated samples. Following (Kynkäänniemi et al., 2019), we use VGG-16 as a feature extractor and StyleGAN2 trained on the FFHQ dataset as the generative model to be assessed. Please see Sec. A.4 for more details.

3. **[Radii]** Recording the distance from each sample to its $k$th nearest neighbour as the radius of its hypersphere, taking $O(n)$ time and space.

4. **[Distance Matrix between $\Phi_r$ and $\Phi_g$]** Calculating pairwise distances between samples of $\Phi_r$ and $\Phi_g$, which consumes $O(n^2)$ time and space.

5. **[P&R]** Calculating P&R ratios, taking $O(n^2)$ time and no extra space for each metric.

In contrast, our efficient P&R can be divided into seven stages:

1. **[Subspace Construction for $\Phi_r$ and $\Phi_g$]** Constructing subspaces of samples for $\Phi_r$ and $\Phi_g$ as required by the approximate $k$-NN algorithm HNSW (Malkov & Yashunin, 2018), taking $O(\log n)$ time and $O(n)$ space for each set.

2. **[Approx. Hubs Identification for $\Phi_r$ and $\Phi_g$]** Computing the approximate hubness value for each sample in $\Phi_r$ and $\Phi_g$ using the approximate $k$-NN algorithm and extracting hubs set $\Phi_r^{hub}$ and $\Phi_g^{hub}$ with $m_r$ and $m_g$ ($m_r < n, m_g < n$) hubs samples respectively using a user-specified threshold $t$, taking $O(m_r)$, $O(m_g)$ time and space for each set, respectively.

3. **[Efficient Distance Matrices]** Calculating pairwise distances for samples between $\Phi_r^{hub}$ and $\Phi_r$, and $\Phi_g^{hub}$ and $\Phi_g$, which consumes $O(m_r n)$ and $O(m_g n)$ time and space, respectively.

4. **[Efficient Sorting]** Sorting the distance matrices as required by the $k$-NN algorithm, which consumes $O(m_r n \log n)$ and $O(m_g n \log n)$ time respectively and no extra space.

5. **[Radii]** Recording the distance from each sample to its $k$th nearest neighbour as the radius of its hypersphere, taking $O(m_r)$ and $O(m_g)$ time and space, respectively.

6. **[Efficient Distance Matrix between $\Phi_r^{hub}$ and $\Phi_g^{hub}$]** Calculating pairwise distances between samples of $\Phi_r^{hub}$ and $\Phi_g^{hub}$, which consumes $O(m_r m_g)$ time and space.

7. **[Efficient P&R]** Calculating P&R ratios, taking $O(m_g^2)$ and $O(m_r^2)$ time and no extra space for each metric.

Theoretically, the proposed eP&R metrics run in $\max(O(m_r n \log n), O(m_g n \log n))$ time and consumes $\max(O(m_r n), O(m_g n))$ space while the original P&R metrics run in $O(n^2 \log n)$ time and consumes $O(n^2)$ space. Since $m_r < n, m_g < n$, the proposed eP&R metrics are far more efficient than the original P&R metrics.

## 5 EXPERIMENTS

### 5.1 EXPERIMENTAL SETUP

**Hardware.** We use a PC with an Intel(R) Core(TM) i7-10875H CPU, an NVIDIA RTX 4090 24GB GPU for small datasets and a GPU node with 2 NVIDIA V100 32GB GPUs for large datasets.

**Datasets.** We use the FFHQ (Kazemi & Sullivan, 2014) dataset containing 70k portrait images, the CelebA-HQ (Karras et al., 2017) dataset containing 30k portrait images and the LSUN (Church, Cat, and Horse) dataset (Yu et al., 2015) containing 120k, 1.5m and 1.5m images of corresponding categories respectively in our experiments.

**Generative Models.** Following (Kynkäänniemi et al., 2019), we test our eP&R metrics with Style-GAN2 (Karras et al., 2020) trained on the FFHQ and LSUN-Cat, LSUN-Church and LSUN-Horse datasets mentioned above. To demonstrate the generalizability of our metrics, we further test them with the other members of the StyleGAN family, including StyleGAN1 (Karras et al., 2019), Style-GAN3 (Karras et al., 2021), VQ-VAE-2 (Razavi et al., 2019) and the Latent Diffusion model (Rombach et al., 2022b) trained on the FFHQ dataset.

**Hyper-parameters.** Unless specified, we follow Kynkäänniemi et al. (2019) and use $k = 3$ in (approximate) $k$-NN algorithms for all P&R, eP&R calculations and hubness-aware sampling, and $t = 3$ as the threshold to extract hubs samples.

Unless specified, we use the FFHQ dataset and a StyleGAN2 model trained on it in our experiments.

### 5.2 EFFICIENT VS. ORIGINAL PRECISION AND RECALL

**Approximation Error.** Our eP&R is an approximation of the original P&R (Kynkäänniemi et al., 2019), which inevitably introduces errors. To demonstrate the validity of our approximation, we record the relative errors $\epsilon = \frac{|x - \hat{x}|}{|x|}$ in Table 1, where $x$ is the original P&R result and $\hat{x}$ is our approximation. It can be observed that our eP&R metrics share almost identical results to the original P&R with very small relative errors around 1%. Please see Appendix A.1 for a comparison with reduced sampling of the original P&R, which further justifies the effectiveness of our metrics.

**Time and Memory Consumption.** We profile the running time and memory consumption to compare the computational efficiency of our eP&R and the original P&R metrics. As Table 2 shows, our eP&R metrics run significantly faster and consume much less memory than the baseline, which justifies our complexity analysis in Sec. 4.3.

### 5.3 ABLATION STUDY

As mentioned in Secs. 4.1 and 4.2, the proposed eP&R metrics consist of three components addressing Observation 4.1, Observation 4.2, and "Efficient Hubs Sample Identification" (approx. $k$-NN) respectively. To show their effectiveness, we conduct an ablation study as shown in Table 3. It can be observed that each of the proposed components contributes to the success of our eP&R metrics.

---

[2]https://github.com/pythonprofilers/memory_profiler/tree/master

Table 1: Approximation errors compared to the original Precision and Recall (P&R) metrics. B.L.: the original P&R metrics as the baseline (Kynkäänniemi et al., 2019). eP&R: our efficient P&R metrics. Error(%): relative error $\epsilon = \frac{|x - \hat{x}|}{|x|}$, where $x$ is the B.L. result and $\hat{x}$ is our eP&R result.

(a) Approximation errors of eP&R computed using StyleGAN2 trained on different datasets.

| | FFHQ | | CelebA-HQ | | LSUN-Church | | LSUN-Cat | | LSUN-Horse | |
|---|---|---|---|---|---|---|---|---|---|---|
| | Precision | Recall | Precision | Recall | Precision | Recall | Precision | Recall | Precision | Recall |
| eP&R | 0.719±0.002 | 0.501±0.002 | 0.709±0.001 | 0.401±0.002 | 0.608±0.002 | 0.362±0.003 | 0.758±0.001 | 0.408±0.003 | 0.782±0.001 | 0.414±0.003 |
| B.L. | 0.716±0.001 | 0.493±0.001 | 0.713±0.001 | 0.394±0.001 | 0.614±0.001 | 0.359±0.002 | 0.766±0.001 | 0.401±0.002 | 0.778±0.001 | 0.409±0.002 |
| Error(%) | 0.4% | 0.2% | 0.6% | 1.7% | 0.9% | 0.8% | 1.0% | 1.7% | 0.5% | 1.2% |

(b) Approximation errors of eP&R calculated using different generative models and the FFHQ dataset ($t = 4$).

| | StyleGAN3 | | StyleGAN1 | | VQ-VAE-2 | | Latent Diffusion | |
|---|---|---|---|---|---|---|---|---|
| | Precision | Recall | Precision | Recall | Precision | Recall | Precision | Recall |
| eP&R | 0.684±0.002 | 0.559±0.001 | 0.724±0.001 | 0.389±0.001 | 0.719±0.001 | 0.163±0.004 | 0.749±0.001 | 0.262±0.002 |
| B.L. | 0.680±0.001 | 0.553±0.001 | 0.719±0.001 | 0.392 ±0.001 | 0.716±0.001 | 0.162±0.001 | 0.743±0.001 | 0.269±0.001 |
| Error(%) | 0.6% | 1.0% | 0.7% | 0.7% | 0.4% | 0.6% | 0.8% | 2.6% |

Table 2: Time and space consumption of our eP&R metrics compared to the original P&R metrics using StyleGAN2 trained on the FFHQ and LSUN-Church datasets respectively. B.L.: the original P&R metrics as the baseline (Kynkäänniemi et al., 2019). eP&R: our efficient P&R metrics. Time (S): serial implementation. Time (P): parallel implementation using CUDA. The profiling items are in one-to-one correspondence with the lists in Sec. 4.3 using Python time() and memory-profiler[2].

(a) FFHQ (70k images).

| Profiling | B.L. | | | Profiling | eP&R | | |
|---|---|---|---|---|---|---|---|
| | Time (S) | Time (P) | Memory | | Time (S) | Time (P) | Memory |
| DMs ($\mathbf{\Phi}_r$, $\mathbf{\Phi}_g$) | 160s | 66s | 15.84 GB | Subspace ($\mathbf{\Phi}_r$, $\mathbf{\Phi}_g$) | 4s | 3s | 3.01 GB |
| | | | | A. hubs ($\mathbf{\Phi}_r^{hub}$, $\mathbf{\Phi}_g^{hub}$) | 2s | 1.2s | – |
| | | | | eDMs | 72s | 32s | 11.23 GB |
| Sorting | 104s | 22s | – | eSorting | 50s | 12s | – |
| Radii | 2.2s | 2.2s | 0.58 GB | Radii | 1.7s | 1.7s | 0.30 GB |
| DM ($\mathbf{\Phi}_r \leftrightarrow \mathbf{\Phi}_g$) | 85s | 34s | 19.24 GB | eDM ($\mathbf{\Phi}_r^{hub} \leftrightarrow \mathbf{\Phi}_g^{hub}$) | 18s | 9s | 8.74 GB |
| P&R | 48s | 28s | – | eP&R | 11s | 6s | – |
| Total/Peak | 399s | 144s | 19.90 GB | Total/Peak | 165s | **75s** | **14.24 GB** |

(b) LSUN-Church (120K images).

| Profiling | B.L. | | | Profiling | eP&R | | |
|---|---|---|---|---|---|---|---|
| | Time (S) | Time (P) | Memory | | Time (S) | Time (P) | Memory |
| DMs ($\mathbf{\Phi}_r$, $\mathbf{\Phi}_g$) | 211s | 110s | 35.24 GB | Subspace ($\mathbf{\Phi}_r$, $\mathbf{\Phi}_g$) | 9s | 7s | 5.40 GB |
| | | | | A. hubs ($\mathbf{\Phi}_r^{hub}$, $\mathbf{\Phi}_g^{hub}$) | 3.3s | 3.3s | – |
| | | | | eDMs | 107s | 48s | 18.83 GB |
| Sorting | 164s | 42s | – | eSorting | 95s | 28s | – |
| Radii | 5s | 5s | 0.81 GB | Radii | 4s | 4s | 0.51 GB |
| DM ($\mathbf{\Phi}_r \leftrightarrow \mathbf{\Phi}_g$) | 143s | 61s | 37.04 GB | eDM ($\mathbf{\Phi}_r^{hub} \leftrightarrow \mathbf{\Phi}_g^{hub}$) | 36s | 15s | 9.54 GB |
| P&R | 90s | 40s | – | eP&R | 17s | 8s | – |
| Total/Peak | 613s | 238s | 37.24 GB | Total/Peak | 269s | **113s** | **25.01 GB** |

## 5.4 CHOICE OF HYPERPARAMETERS

The proposed eP&R metrics have two hyperparameters: i) $k$ used by the (approximate) $k$-NN algorithm; and ii) threshold $t$ used to identify hubs samples.

**Choice of number of nearest neighbours $k$.** As Table 4 shows, it can be observed that improvements of our eP&R metrics are stable under different choices of $k$. Therefore, without loss of generality, we use $k = 3$ following Kynkäänniemi et al. (2019).

**Choice of threshold $t$.** As Table 5 shows, our eP&R metrics introduce a trade-off between error and efficiency with $t$: the higher $t$, the more efficient our metrics but at the cost of higher errors. Thus, in our experiments, we strike a balance by using $t = 3$ for (FFHQ, StyleGAN2) combination.

Table 5: Choice of threshold $t$.

| $t$ | Precision | Recall | Time (P) | Memory |
|---|---|---|---|---|
| 1 | 0.717±0.001 | 0.500±0.002 | 116s | 16.34 GB |
| 2 | 0.718±0.001 | 0.501±0.002 | 95s | 15.27 GB |
| 3 | 0.719±0.002 | 0.501±0.001 | 75s | 14.21 GB |
| 4 | 0.726±0.002 | 0.507±0.001 | 68s | 13.10 GB |
| 5 | 0.730±0.001 | 0.515±0.002 | 63s | 12.28 GB |
| B.L. | 0.716±0.001 | 0.493±0.001 | 144s | 19.90GB |

Table 3: Ablation study. Alg. Variations: variants of our metrics. Time (P): parallel implementation using CUDA. P&R: original P&R metrics (Kynkäänniemi et al., 2019). Ob. 4.1: we replace the $\Phi_g$ in precision calculation and $\Phi_r$ in recall calculation with their hubs versions $\Phi_g^{hub}$ and $\Phi_r^{hub}$ respectively (Eq. 1). Ob. 4.2: we replace the $\Phi_r$ in precision calculation and $\Phi_g$ in recall calculation with their hubs versions $\Phi_r^{hub}$ and $\Phi_g^{hub}$ respectively (Eq. 1). eP&R: our efficient P&R metrics, which uses Ob. 4.1, 4.2 and "Efficient Hubs Sample Identification" (approximate $k$-NN) together. *: when used alone, (2)(3) cannot save time and space as they still require the full distance matrices.

| Alg. Variations | Precision | Recall | Time (P) | Memory |
|---|---|---|---|---|
| (1) P&R (Original) | 0.716±0.001 | 0.493±0.001 | 144s | 19.90 GB |
| (2) P&R + Ob. 4.1 | 0.715±0.005 | 0.497±0.004 | 138s* | 19.32* GB |
| (3) P&R + Ob. 4.2 | 0.708±0.005 | 0.501±0.005 | 140s* | 19.31* GB |
| (4) P&R + Ob. 4.1, 4.2 | 0.719±0.002 | 0.494±0.001 | 104s | 15.84 GB |
| (5) eP&R (Ours) | 0.719±0.002 | 0.501±0.001 | **75s** | **14.21 GB** |

Table 4: Choice of $k$ for the $k$-NN algorithm used in our eP&R metric. B.L.: the original P&R metrics as the baseline (Kynkäänniemi et al., 2019). eP&R: our efficient P&R metrics. Time (S): serial implementation. Time (P): parallel implementation using CUDA.

| $k$ | B.L. | | | | eP&R | | | |
|---|---|---|---|---|---|---|---|---|
| | Precision | Recall | Time (P) | Memory | Precision | Recall | Time (P) | Memory |
| 3 | 0.716±0.001 | 0.493±0.001 | 144s | 19.90 GB | 0.719±0.001 | 0.494±0.001 | 75s | 14.21 GB |
| 5 | 0.814±0.001 | 0.614±0.001 | 150s | 19.90 GB | 0.813±0.002 | 0.615±0.002 | 73s | 13.71 GB |
| 7 | 0.865±0.001 | 0.683±0.001 | 144s | 19.91 GB | 0.868±0.004 | 0.689±0.002 | 73s | 13.51 GB |
| 9 | 0.893±0.001 | 0.730±0.001 | 147s | 19.91 GB | 0.899±0.006 | 0.737±0.002 | 72s | 13.39 GB |
| 10 | 0.907±0.001 | 0.758±0.001 | 147s | 19.91 GB | 0.915±0.006 | 0.767±0.002 | 71s | 13.39 GB |

## 5.5 ROBUSTNESS AGAINST THE TRUNCATION TRICK

The truncation trick is a widely used technique that improves GAN sample quality by truncating the latent vector $z$ fed into the generator (Brock et al., 2019; Karras et al., 2019; 2020). As Table 6 shows, our eP&R metrics are robust against the truncation trick with $\phi = 0.5, 0.7$, where $\phi = 0.7$ is the recommended value.

Table 6: Robustness against the truncation trick (Karras et al., 2019). We calculate the metrics using StyleGAN2 trained on the FFHQ dataset and $t = 4$. Please note that $\phi = 0.7$ is the recommended value Karras et al. (2019; 2020) for the truncation trick and $\phi = 1.0$ means no truncation is applied at all. B.L.: the original P&R metrics as the baseline (Kynkäänniemi et al., 2019). eP&R: our efficient P&R metrics. Error(%): relative error $\epsilon = \frac{|x-\hat{x}|}{|x|}$, where $x$ is the B.L. result and $\hat{x}$ is our eP&R result. We did not include Time and Memory costs are the truncation trick does not affect the number of samples, hence consuming the same amount of time and memory.

| | $\phi = 0.5$ | | $\phi = 0.7$ | | $\phi = 1.0$ | |
|---|---|---|---|---|---|---|
| | Precision | Recall | Precision | Recall | Precision | Recall |
| eP&R | 0.932±0.002 | 0.089±0.002 | 0.890±0.002 | 0.297±0.002 | 0.714±0.002 | 0.493±0.001 |
| B.L. | 0.935±0.001 | 0.101±0.001 | 0.885±0.001 | 0.308±0.001 | 0.716±0.001 | 0.493±0.001 |

## 6 CONCLUSION

In conclusion, we have proposed efficient precision and recall (eP&R) metrics that provide almost identical results as the original P&R metrics but with much lower computational costs. By identifying and removing redundancies in P&R computation through hubness-aware sampling and approximate $k$-NN methods, we have developed a highly efficient yet accurate approach to evaluating generative models. Extensive experiments demonstrate the effectiveness of our eP&R metrics. Going forward, eP&R provides an important step towards feasible and insightful assessment of state-of-the-art generative models trained on massive datasets. We believe eP&R can enable more rapid progress in this exciting field.

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

# A    ADDITIONAL EXPERIMENTS

## A.1    COMPARISON WITH REDUCED SAMPLING

To further demonstrate the superiority of our eP&R metrics, we compare them with another baseline of reduced sampling, *i.e.*, instead of using the full dataset, we randomly sample a subset from it and use a reduced number of generated samples to calculate P&R accordingly. As Table 7 shows, our method provides much more accurate P&R results given the same number of samples, demonstrating the superiority of our metrics.

Table 7: Comparison with reduced sampling. # of Spls: number of samples. R.S.: reduced sampling, *i.e.*, instead of using the full dataset, we randomly sample a subset from it and use a reduced number of generated samples to calculate P&R accordingly. eP&R: our efficient P&R metrics. Time (S): serial implementation. Time (P): parallel implementation using CUDA. The last row shows the results of the original P&R as a reference.

| # of Spls (%) | R.S. | | | | eP&R | | | |
| | Precision | Recall | Time (P) | Memory | Precision | Recall | Time (P) | Memory |
|---|---|---|---|---|---|---|---|---|
| 72.50 | 0.724±0.002 | 0.511±0.002 | 108s | 13.12 GB | 0.717±0.001 | 0.500±0.002 | 116s | 16.34 GB |
| 52.24 | 0.730±0.002 | 0.522±0.002 | 73s | 12.23 GB | 0.718±0.001 | 0.501±0.002 | 95s | 15.27 GB |
| 38.21 | 0.734±0.005 | 0.533±0.005 | 49s | 8.47 GB | 0.719±0.002 | 0.501±0.001 | 75s | 14.21 GB |
| 28.48 | 0.742±0.005 | 0.540±0.005 | 34s | 6.22 GB | 0.726±0.002 | 0.507±0.001 | 68s | 13.10 GB |
| 21.65 | 0.743±0.005 | 0.551±0.006 | 25s | 5.01 GB | 0.730±0.001 | 0.515±0.002 | 63s | 12.28 GB |
| P&R (70k) | 0.716±0.001 | 0.493±0.001 | 144s | 19.90GB | - | - | - | - |

## A.2    TIME AND SPACE CONSUMPTION FOR LARGE DATASETS

Due to hardware limitations, we have to perform matrix tiling when calculating P&R and eP&R on large datasets which splits a given matrix into tiles (submatrices) that can fit into GPU memory. However, this introduces additional overheads and is not desirable (Table 8), which further justifies our motivation to design efficient evaluation metrics for generative models. Nevertheless, we show the results of our eP&R metrics on the LSUN-Horse dataset containing 1.5m images in Table 9. It can be observed that our metrics still save a lot of time when matrix tiling is used.

Table 8: Time costs when matrix tiling is used. The experiments are conducted using the FFHQ dataset and a StyleGAN2 model trained on it.

| # of Tiles | 1 (no tiling) | 2 | 5 | 10 | 50 | 100 |
|---|---|---|---|---|---|---|
| Time (P) | 144s | 146s | 148s | 150s | 174s | 192s |

Table 9: Time and space consumption of our eP&R metrics compared to the original P&R metrics using StyleGAN2 trained on the LSUN-Horse dataset. B.L.: the original P&R metrics as the baseline (Kynkäänniemi et al., 2019). eP&R: our efficient P&R metrics. Time (S): serial implementation. Time (P): parallel implementation using CUDA. The profiling items are in one-to-one correspondence with the stages listed in Sec. 4.3.

| Profiling | B.L. | | Profiling | eP&R | |
| | Time (S) | Time (P) | | Time (S) | Time (P) |
|---|---|---|---|---|---|
| | | | Subspace $(\mathbf{\Phi}_r, \mathbf{\Phi}_g)$ | 8min | 3min |
| DMs $(\mathbf{\Phi}_r, \mathbf{\Phi}_g)$ | 12h02min | 54min | A. hubs $(\mathbf{\Phi}_r^{hub}, \mathbf{\Phi}_g^{hub})$ | 2min | 1min |
| | | | eDMs | 5h56min | 26min |
| Sorting | 2h56min | 22min | eSorting | 1h20min | 11min |
| Radii | 1min | 1min | Radii | 50s | 50s |
| DM $(\mathbf{\Phi}_r \leftrightarrow \mathbf{\Phi}_g)$ | 6h30min | 34min | eDM $(\mathbf{\Phi}_r^{hub} \leftrightarrow \mathbf{\Phi}_g^{hub})$ | 1h27min | 8min |
| P&R | 2h12min | 17min | eP&R | 56min | 4min |
| Total | 23h45min | 2h10min | Total | 9h50min | **64min** |

### A.3 P&R Curves

We follow (Kynkäänniemi et al., 2019) and include the original P&R (baseline) and our eP&R curves against the parameter of the truncation trick in Fig. 3. The results show that our method approximates the original P&R curves well on both FFHQ and LSUN-Church datasets.

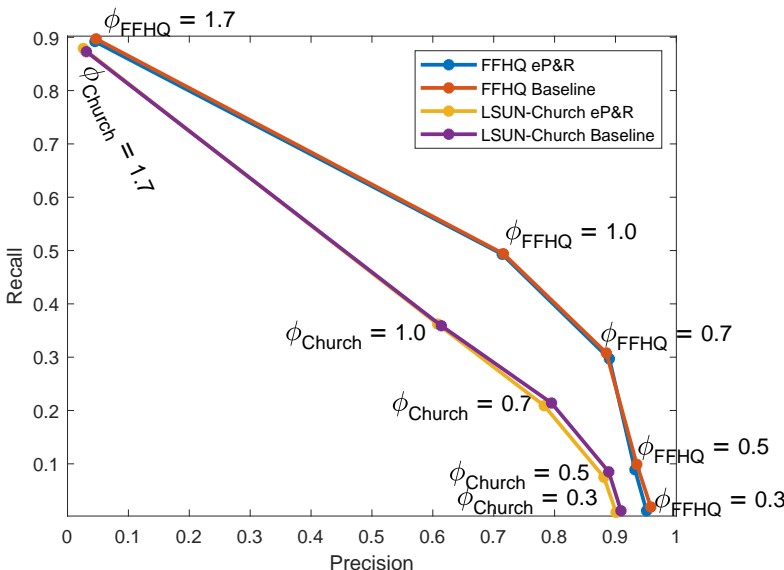

Figure 3: Original P&R (baseline) and our eP&R curves on the FFHQ and LSUN-Church datasets.

### A.4 Insensitivity to Group Split Points

As shown in Table 10, the ratios of binary scores are similar for each hubness value on the FFHQ dataset, which validates the insensitivity of the choice of group split points for Observation 4.1 and Fig. 1 in the main paper.

Similarly, as shown in Table 11, the ratios of hubness samples increase quickly to 1 with the increase of $|\phi'|$ on the FFHQ dataset, which validates the insensitivity of the choice of group split points for Observation 4.2 and Fig. 2.

We show the same conclusions hold on the LSUN-Church dataset as well (Table 12 and Table 13).

### A.5 Justification of pairwise distance calculation between $\Phi_r^{hub}$ and $\Phi_r$

As Table 14 shows, we calculate the pairwise distances between $\Phi_r^{hub}$ and $\Phi_r$ as it provides lower approximation errors than calculating pairwise distances for samples in $\Phi_r^{hub}$. We conjecture the reason is that $\Phi_r^{hub}$ is much sparser than $\Phi_r$ and thus the pairwise distances for samples in it will be much larger than those of the original P&R, resulting in much larger $k$-NN hyperspheres that increase the approximation error. The same conclusion holds for $\Phi_g^{hub}$ and $\Phi_g$.

### A.6 Additional experiments about the choice of $t$

As Table 5 shows, our eP&R metrics introduce a trade-off between error and efficiency with $t$: the higher $t$, the more efficient our metrics but at the cost of higher errors. In Table 15, we provide more experimental results on the choices of $t$.

## B Limitations and Future Work

Although effective and efficient, the proposed eP&R metrics are not fully optimized. One area for improvement is in Stage 3 (Efficient Distance Matrices), which currently calculates pairwise

Table 10: Insensitivity to group split points for Observation 4.1 and Fig. 1 in the main paper (FFHQ). Hub. Value: hubness value, B.S.: binary score.

| Hub. Value | 1 | 2 | 3 | 4 | 5 | 6 | 7 | 8 | 9 | 10 | 11 | 12 |
|---|---|---|---|---|---|---|---|---|---|---|---|---|
| B.S. = 1 | 9038 | 7489 | 5795 | 4579 | 3525 | 2741 | 2188 | 1842 | 1474 | 1244 | 931 | 818 |
| All Samples | 13382 | 11292 | 8615 | 6815 | 5250 | 4158 | 3235 | 2664 | 2190 | 1791 | 1389 | 1231 |
| Ratio | 0.675 | 0.663 | 0.673 | 0.672 | 0.671 | 0.659 | 0.676 | 0.691 | 0.673 | 0.695 | 0.67 | 0.665 |

| Hub. Value | 13 | 14 | 15 | 16 | 17 | 18 | 19 | 20 | 21 | 22 | 23 | 24 |
|---|---|---|---|---|---|---|---|---|---|---|---|---|
| B.S. = 1 | 679 | 593 | 508 | 425 | 361 | 305 | 272 | 246 | 200 | 177 | 153 | 160 |
| All Samples | 1046 | 883 | 776 | 640 | 543 | 469 | 409 | 361 | 291 | 264 | 232 | 230 |
| Ratio | 0.649 | 0.672 | 0.655 | 0.664 | 0.665 | 0.65 | 0.665 | 0.681 | 0.687 | 0.67 | 0.659 | 0.696 |

(a) All 70k images in the FFHQ dataset

| Hub. Value | 1 | 2 | 3 | 4 | 5 | 6 | 7 | 8 | 9 | 10 | 11 | 12 |
|---|---|---|---|---|---|---|---|---|---|---|---|---|
| B.S.e = 1 | 10030 | 9150 | 7375 | 5858 | 4555 | 3664 | 2952 | 2266 | 1848 | 1515 | 1358 | 1120 |
| All Samples | 12083 | 10988 | 8833 | 7015 | 5452 | 4402 | 3561 | 2743 | 2237 | 1835 | 1624 | 1344 |
| Ratio | 0.83 | 0.833 | 0.835 | 0.835 | 0.835 | 0.832 | 0.829 | 0.826 | 0.826 | 0.826 | 0.836 | 0.833 |

| Hub. Value | 12 | 13 | 14 | 15 | 16 | 17 | 18 | 19 | 20 | 21 | 22 | 23 | 24 |
|---|---|---|---|---|---|---|---|---|---|---|---|---|---|
| B.S.e = 1 | 1120 | 911 | 800 | 646 | 522 | 456 | 374 | 350 | 306 | 261 | 228 | 187 | 178 |
| All Samples | 1117 | 992 | 762 | 635 | 539 | 467 | 420 | 379 | 317 | 277 | 217 | 208 | |
| Ratio | 0.816 | 0.806 | 0.848 | 0.822 | 0.846 | 0.801 | 0.833 | 0.807 | 0.823 | 0.823 | 0.862 | 0.856 | |

(b) 70k images generated by StyleGAN2

Table 11: Insensitivity to group split points for Observation 4.2 and Fig. 2 in the main paper (FFHQ).

| $\phi'$ | 0 | 1 | 2 | 3 | 4 | 5 | $\geq 6$ |
|---|---|---|---|---|---|---|---|
| Hubness | 11922 | 8625 | 6290 | 4540 | 3516 | 2741 | 10818 |
| Non-hubness | 13254 | 9089 | 6519 | 4598 | 3537 | 2749 | 10818 |
| Ratio | 0.9 | 0.949 | 0.965 | 0.987 | 0.994 | 0.997 | 1.000 |

(a) All 70k images in the FFHQ dataset

| $\phi'$ | 0 | 1 | 2 | 3 | 4 | $\geq 5$ |
|---|---|---|---|---|---|---|
| Hubness | 11792 | 6596 | 4136 | 2745 | 1860 | 5531 |
| Non-hubness | 13482 | 7210 | 4328 | 2788 | 1874 | 5531 |
| Ratio | 0.875 | 0.915 | 0.956 | 0.985 | 0.992 | 1.000 |

(b) 70k images generated by StyleGAN2

distances between hub samples of one set and all samples of the other set to compute radii. A significant amount of time is spent on this step. We could optimize this by utilizing the subspace constructed by the approximate $k$-NN algorithms. Instead of comparing hubs to the full set, we would only need to calculate distances between hubs and samples within the relevant subspace of the other set. This would allow us to find radii much more quickly. While the current metrics are fast and accurate, optimizations like these could push the efficiency even higher without sacrificing effectiveness. We therefore see continued refinement of the eP&R metrics represents an exciting opportunity for future work.

## C    A BRIEF INTRODUCTION TO APPROXIMATE K-NN ALGORITHM

The $k$-nearest neighbors ($k$-NN) algorithm is a popular machine learning method for classification and regression. Given a new data point, it finds the $k$ closest training examples based on a distance metric like Euclidean distance. A major limitation of $k$-NN is that it requires computing the distance between the new point and all points in the training set, which can be slow for large datasets.

Approximate $k$-NN algorithms are techniques that try to speed up neighbor search by sacrificing some accuracy. The key idea is to avoid exhaustively calculating distances to all points. Some common approaches include:

Table 12: Insensitivity to group split points for Observation 4.1 and Fig. 1 in the main paper (LSUN-Church). Hub. Value: hubness value, B.S.: binary score.

| Hubness Value | 1 | 2 | 3 | 4 | 5 | 6 | 7 | 8 | 9 |
|---|---|---|---|---|---|---|---|---|---|
| Binary Score = 1 | 51045 | 17414 | 6653 | 2906 | 1413 | 658 | 355 | 202 | 108 |
| All Samples | 75650 | 25836 | 9866 | 4296 | 2074 | 1006 | 562 | 291 | 153 |
| Ratio | 0.675 | 0.674 | 0.674 | 0.676 | 0.681 | 0.654 | 0.631 | 0.694 | 0.708 |

(a) All 120k images in the LSUN-Church dataset

| Hubness Value | 1 | 2 | 3 | 4 | 5 | 6 | 7 | 8 | 9 | 10 |
|---|---|---|---|---|---|---|---|---|---|---|
| Binary Score = 1 | 24459 | 4658 | 1081 | 405 | 161 | 71 | 35 | 19 | 9 | 6 |
| All Samples | 55401 | 10564 | 2385 | 957 | 371 | 154 | 81 | 38 | 21 | 12 |
| Ratio | 0.441 | 0.441 | 0.453 | 0.423 | 0.434 | 0.461 | 0.432 | 0.5 | 0.429 | 0.5 |

(b) 100k images generated by StyleGAN2

Table 13: Insensitivity to group split points for Observation 4.2 and Fig. 2 in the main paper (LSUN-Church).

| $\phi'$ | 0 | 1 | 2 | 3 | 4 | 5 | $\geq 6$ |
|---|---|---|---|---|---|---|---|
| Hubness | 29986 | 18482 | 12819 | 9106 | 6749 | 5037 | 20441 |
| Non-hubness | 32669 | 19615 | 13346 | 9213 | 6773 | 5040 | 20441 |
| Ratio | 0.918 | 0.942 | 0.961 | 0.988 | 0.996 | 0.999 | 1 |

(a) All 120k images in the LSUN-Church dataset

| $\phi'$ | 0 | 1 | 2 | 3 | 4 | 5 | 6 | $\geq 7$ |
|---|---|---|---|---|---|---|---|---|
| Hubness | 15731 | 7063 | 3984 | 2405 | 1505 | 963 | 728 | 2007 |
| Non-hubness | 17590 | 7821 | 4158 | 2443 | 1513 | 965 | 729 | 2007 |
| Ratio | 0.894 | 0.903 | 0.958 | 0.984 | 0.995 | 0.998 | 0.999 | 1 |

(b) 100k images generated by StyleGAN2

Table 14: Justification of pairwise distance calculation between $\Phi_r^{hub}$ and $\Phi_r$. P: Precision; R: Recall. Error(%): relative error $\epsilon = \frac{|x - \hat{x}|}{|x|}$, where $x$ denotes the baseline precision 0.716 and recall 0.493 of the original P&R metric.

| $t$ | Pairwise distance between $\Phi_r^{hub}$, $\Phi_r^{hub}$ | | | | Pairwise distance between $\Phi_r^{hub}$, $\Phi_r$ | | | |
|---|---|---|---|---|---|---|---|---|
| | P | R | P Error(%) | R Error(%) | P | R | P Error(%) | R Error(%) |
| 1 | 0.713 | 0.484 | 0.4 | 1.8 | 0.717 | 0.500 | 0.3 | 0.0 |
| 2 | 0.723 | 0.506 | 1.0 | 2.6 | 0.718 | 0.501 | 0.3 | 0.0 |
| 3 | 0.746 | 0.534 | 4.2 | 8.3 | 0.719 | 0.501 | 0.4 | 0.2 |
| 4 | 0.768 | 0.562 | 7.3 | 14.0 | 0.726 | 0.507 | 1.6 | 0.6 |
| 5 | 0.787 | 0.588 | 9.9 | 19.3 | 0.730 | 0.515 | 1.9 | 0.6 |

Table 15: The eP&R scores with different threshold $t$. Error(%): relative error $\epsilon = \frac{|x-\hat{x}|}{|x|}$

| | | Hubness | | Error(%) | | |
|---|---|---|---|---|---|---|
| $t$ | Percent(%) | Precision | Recall | Precision | Recall | Mean |
| 1 | 72.50±0.01 | 0.718±0.001 | 0.494±0.002 | 0.3 | 0.0 | 0.1 |
| 2 | 52.24±0.04 | 0.718±0.001 | 0.494±0.002 | 0.3 | 0.0 | 0.1 |
| 3 | 38.21±0.04 | 0.719±0.002 | 0.494±0.001 | 0.4 | 0.2 | 0.3 |
| 4 | 28.48±0.04 | 0.726±0.002 | 0.496±0.001 | 1.6 | 0.6 | 1.1 |
| 5 | 21.65±0.04 | 0.730±0.001 | 0.496±0.002 | 1.9 | 0.6 | 1.3 |
| 6 | 16.64±0.02 | 0.732±0.002 | 0.497±0.003 | 2.2 | 0.6 | 1.4 |
| 7 | 12.99±0.02 | 0.739±0.002 | 0.498±0.003 | 3.2 | 2.6 | 2.8 |
| 8 | 10.22±0.01 | 0.747±0.002 | 0.509±0.003 | 4.3 | 4.3 | 4.3 |
| 9 | 8.15±0.03 | 0.747±0.003 | 0.509±0.004 | 5.5 | 8.1 | 6.8 |
| 10 | 6.55±0.01 | 0.748±0.004 | 0.517±0.003 | 9.9 | 8.4 | 8.4 |
| B.L. | — | 0.716±0.001 | 0.493±0.001 | — | — | — |

- Tree-based data structures like $kd$-trees that allow efficient searching of nearest points without checking all data.
- Hashing techniques that map similar points to the same buckets, narrowing the search.
- Dimensionality reduction methods like random projections that can compress data while preserving relative distances.
- Graph-based algorithms that connect neighboring points then traverse the graph instead of computing all distances.
- Sampling/filtering methods that find candidates in subsections of data.

The tradeoff is between accuracy and speed. Approximate methods may miss some true nearest neighbors, but can query large datasets much more efficiently. Performance gains allow $k$-NN to scale better to big data. Appropriate techniques depend on factors like data size, dimension, and desired accuracy. We refer interested audiences to (Aumüller et al., 2017; Li et al., 2019; Shimomura et al., 2021; Wang et al., 2021) for more details.

## D    ILLUSTRATION FIGURE AND RELEVANT DISCUSSIONS FOR VALID $\phi'$

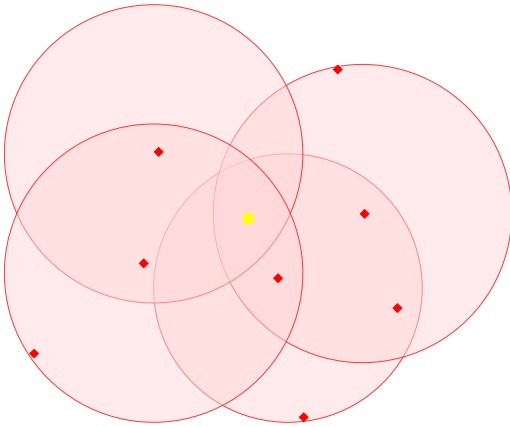

Figure 4: Illustration of valid $\phi'$. $\phi$ is represented by a yellow cube and $\phi' \in \Phi$ set are represented by red rhombuses.

As Fig. 4 shows, by "the times a sample is included in the $k$-NN hypersphere of a sample of the other distribution, *i.e.*, valid $\phi'$", we count the number of times $\phi$ (yellow cube) is within the $k$-NN hypersphere of $\phi' \in \Phi$ (red rhombuses).

