# OpenReview forum: "Efficient Precision and Recall Metrics for Assessing Generative Models using Hubness-aware Sampling"
_ICLR.cc/2024/Conference — Submitted to ICLR 2024_

### Official Review · Reviewer_KcFU · 2023-10-31

**Soundness:** 2 fair
**Presentation:** 2 fair
**Contribution:** 2 fair
**Rating:** 5
**Confidence:** 2

**Summary:**

Drawing from the two redundant problems of Kynkäänniemi et al. (2019) that employing representative subsets of generative and real samples would give the similar results as standard Precision and Recall (P&R) ratio, and empirical observations of the dataset that 1) samples of similar hubness values have the similar ratios of 1 vs. 0 in P&R, and 2) phi^prime with high hubness values are enough for manifold identification, the authors propose a method using subsets of generative and real samples with certain hubness criterion in conjunction with approx. k-NN to reduce time and space complexity.

**Strengths:**

The observations in Fig. 1 and 2 are intriguing.
The authors dissected ratio and manifold identification as separate components and conducted well-supported experiments.
The results are promising.

**Weaknesses:**

The observations in Fig. 1 and 2 are highly empirical while they serve as necessary foundations of the method.

**Questions:**

The description of Fig. 2 is confusing. For example, "Hubness" and "non_hubness" are only explained the the main text not in the description of the figure. And I cannot understand "the times a sample is included in the k-NN hypersphere of a sample of the other distribution, i.e., valid φ′ (FFHQ)".
Please add theoretical analysis of the interesting observations in section 4.2.
A brief introduction of approximate k-NN method would be helpful (but since I am not an expert in this filed, it depends on you).
Since the observations are highly empirical, could you add more experiments about t choice (experiments in table 5).

---

> ### Author Response · Authors · 2023-11-13
> **[Friendly reminder] There might be some technical issue with openreview submission system, as the review doesn't seem to be for our paper**
>
> Dear Reviewer KcFU,
>
> We would like to thank you for taking the time to review our paper. However, it appears there may have been a technical issue with the openreview system as the attached review does not seem to be for our submitted work. We would be very grateful if you could please take a look again and upload the intended review for our paper.
>
> We look forward to receiving your feedback on our work and addressing any concerns you may have. Please do not hesitate to contact us if you need any clarification or have any other questions. We greatly appreciate you taking the time to provide your expert assessment of our research.
>
> Thank you in advance and best regards,
>
> The authors

---

> > ### Comment · Reviewer_KcFU · 2023-11-13
> > **Updated**
> >
> > I genuinely apologize for the confusion.

---

> ### Author Response · Authors · 2023-11-19
> **Authors' Response**
>
> Thank you for your comments and we hope our responses address your concerns.
>
> > **Q1:** The observations in Fig. 1 and 2 are highly empirical while they serve as necessary foundations of the method.
>
> **Re:** As we responded to Reviewer SeVE (Q3),
> we acknowledge the comment that our contributions are currently empirical in nature. While developing theoretical understanding is important, we believe there is also value in empirical studies exposing real-world phenomena about deep learning systems and their applications. As evidenced by the rapid practical progress in deep learning, empirical findings have driven advancement even when lacking theoretical grounding initially.
> However, we agree formalizing the observed trends is an interesting direction and leave it to future work.
>
> > **Q2:** The description of Fig. 2 is confusing. For example, "Hubness" and "non_hubness" are only explained the the main text not in the description of the figure. And I cannot understand "the times a sample is included in the k-NN hypersphere of a sample of the other distribution, i.e., valid φ′ (FFHQ)". Please add theoretical analysis of the interesting observations in section 4.2. A brief introduction of approximate k-NN method would be helpful (but since I am not an expert in this filed, it depends on you). Since the observations are highly empirical, could you add more experiments about t choice (experiments in table 5).
>
> **Re:** Thank you for your suggestions. We have i) added an explanation of "Hubness" and "non_hubness" in the figure description; ii) added a brief introduction of the approximate k-NN algorithm in Sec. C (appendix) of the revised paper; iii) added more experiments about $t$ choices in Table 15 Sec. A.6 (appendix) of the revised paper.
>
> For the meaning of "the times a sample is included in the k-NN hypersphere of a sample of the other distribution, i.e., valid φ′ (FFHQ)", we have included an illustration figure and relevant discussions in Sec. D (appendix) of the revised paper. Unfortunately, we are unable to provide a theoretical analysis at this stage, as the hubness phenomenon is still largely an empirical observation, with limited theoretical understanding developed so far.

---

> > ### Comment · Reviewer_KcFU · 2023-11-19
> >
> > Thank you for response and explaining. As you said you can't give a theoretical basis for your claims, and there's no guarantee these properties apply to every dataset. Using this method might affect metric calculations, potentially misleading researchers. Since you didn't solve my concern, my score remains unchanged. I appreciate your efforts and the new findings.

---

### Official Review · Reviewer_SeVE · 2023-11-01

**Soundness:** 3 good
**Presentation:** 3 good
**Contribution:** 3 good
**Rating:** 6
**Confidence:** 4

**Summary:**

The paper addresses the problem of efficiently assessing generative models on their precision and recall. Intuitively, precision of a generative model measures the quality of samples produced, and recall measures the coverage or diversity of the samples. Unlike scalar evaluation metrics like inception score and FID, computing precision and recall is much more computationally intensive (quadratic complexity in samples, as opposed to linear) because of the need for measuring pairwise distances between the samples. This paper exploits the "hubness" property of high-dimensional datasets to speed up the computation of precision and recall.

To estimate precision and recall of a model with output distribution $\hat p$ against a true distribution $p$, we need to estimate how much of $p$  is covered by $\hat p$ and vice-versa. A popular way to do this (proposed by Kynka¨anniemi et al 2019) is by measuring how many samples of $p$ fall within the support of $\hat p$ where the support is approximated by a union of hyperspheres centered around samples from $\hat p$ with radii being the distance to kth nearest neighbors. (There are other ways to estimate precision and recall, for example, Simon et al 2019 use a discriminator to classify samples from both distributions, but this paper focuses on the Kynka¨anniemi et al method.) The hubness phenomenon results in a few samples from both $p$ and $\hat p$ to be the most popular nearest neighbors to almost all points. This paper exploits this by first using a linear time algorithm to find these "hubs" and then use these to compute P&R.
Through experiments, the paper demonstrates the savings in compute and storage, as compared to Kynka¨anniemi et al P&R evaluation.

**Strengths:**

- The paper well written, and explains the proposed method clearly.
- The experiments convincingly demonstrate savings on compute time and storage for real world datasets, across a variety of model architectures.
- The ablation study and the experiment on robustness against the truncation trick are a nice addition to the experiments section.

**Weaknesses:**

- The proposed speedup is specific to one particular way of P&R estimation i.e. using the Kynka¨anniemi et al 2019 method based on nearest neighbors. This method only gives two scalar values corresponding to P and R. In contrast, Simon et al. ICML 2019 method gives the whole PR curve.
- The proposed method seems to work well when the P&R values are "reasonably good" i.e., away from 0 and 1. It is not clear how well the method works in corner cases. It would be good to check this with toy experiments on high dimensional Gaussian mixtures for which P&R take corner values close to 0 and 1 also.
- Although the experiments are convincing, no theoretical guarantees are provided that bound the approximation error.

**Questions:**

- As I stated above, it would be interesting to see how well the proposed approximations hold up on models with relatively poor P&R, not just good models. It would be good to check this with toy experiments on high dimensional Gaussian mixtures for which P&R take corner values close to 0 and 1 also.

---

> ### Author Response · Authors · 2023-11-19
> **Authors' Response**
>
> Thank you for your comments and we hope our responses address your concerns.
>
> > **Q1:** The proposed speedup is specific to one particular way of P&R estimation i.e. using the Kynka¨anniemi et al 2019 method based on nearest neighbors. This method only gives two scalar values corresponding to P and R. In contrast, Simon et al. ICML 2019 method gives the whole PR curve.
>
> **Re:** Thank you for your suggestion, since our method is based on [1], we follow it and have included the P&R curves against the parameter of the truncation trick in Fig. 3, Sec. A.3 (appendix) of the revised paper.
> The results show that our method approximates the original P&R curves well on both FFHQ and LSUN-Church datasets.
>
> [1] Kynkäänniemi, T., Karras, T., Laine, S., Lehtinen, J. and Aila, T., 2019. Improved precision and recall metric for assessing generative models. Advances in Neural Information Processing Systems, 32.
>
> > **Q2:** The proposed method seems to work well when the P&R values are "reasonably good" i.e., away from 0 and 1. It is not clear how well the method works in corner cases. It would be good to check this with toy experiments on high dimensional Gaussian mixtures for which P&R take corner values close to 0 and 1 also.
>
> **Re:** Good question! As shown in the P&R curves figure newly included in Fig. 3, Sec. A.3 (appendix) of the revised paper, our method also works well in corner cases.
>
>
> > **Q3:** Although the experiments are convincing, no theoretical guarantees are provided that bound the approximation error.
>
> **Re:** We acknowledge the comment that our contributions are currently empirical in nature. While developing theoretical understanding is important, we believe there is also value in empirical studies exposing real-world phenomena about deep learning systems and their applications. As evidenced by the rapid practical progress in deep learning, empirical findings have driven advancement even when lacking theoretical grounding initially.
> However, we agree formalizing the observed trends is an interesting direction and leave it to future work.

---

### Official Review · Reviewer_ctE8 · 2023-11-01

**Soundness:** 3 good
**Presentation:** 3 good
**Contribution:** 2 fair
**Rating:** 5
**Confidence:** 4

**Summary:**

The paper presents efficient Precision and Recall (eP&R) metrics for evaluating deep generative models trained on large-scale datasets, which provide nearly identical results to the original P&R metrics with less computational costs. The authors propose a hubness-aware sampling method to remove two kinds of calculating redundancy in original P&R metrics. Besides, the efficiency of eP&R is further improved by adopting approximate k-NN methods. Experiments conducted confirm the effectiveness and generalizability of the eP&R metrics.

**Strengths:**

1. This work proposes efficient precision and recall (eP&R) metrics for assessing generative models to approximate results as the original P&R metrics with lower consumption in time and space. Specifically, eP&R metrics reduce time complexity from $O(n^2logn)$ to $O(mnlogn)$ and reduce space complexity from $O(n^2)$ to $O(mn)$, where $m$ is less than $n$.
2. In addition, an approximate k-NN algorithm is employed for the identification of hub samples to further improve the efficiency of eP&R metrics.

**Weaknesses:**

1. The authors indicate in Sec. 4.3 that the numbers of hub samples, i.e. $m_r$ and $m_g$ are far less than the number of samples of original sets, i.e. $n$. However, the specific conditions for the validity of this conclusion are not provided. From the experimental results in Table 2, the ratio of $O(m_r)$ or $O(m_g)$ to $O(n)$ is about 0.6, which is not consistent with the statement $m_r \ll n, m_g \ll n$.
2. In Observation 4.1 and Figure 1, the authors roughly divide hubness values into three groups and claim that samples with similar hubness values are effective representative samples in P&R ratio calculation, which lacks generality and specific analysis. Further illustration is needed to explain why the hubness value split points are chosen as 12 and 24, and whether this observation holds in many other datasets. Observation 4.2 and Figure 2 have the same issue.
3. In Sec. 4.2, the authors point out the insensitivity of hubness-aware sampling to exact k-nearest neighbor (k-NN) results, which might be confusing since in Table 4, the Precision and Recall change greatly when k is taken from 3 to 10. Therefore, specific mathematical descriptions are required to substantiate this viewpoint.
4. The font size of the annotations in Figure 1 and Figure 2 is too small to identify clearly. Besides, the explanation for (a) in Figure 2 is unclear, which can be directly replaced by 'hubness' and 'non-hubness'.

**Questions:**

1. In Sec. 4.3 in the third stage of complexity analysis for eP&R, why calculating pairwise distances for samples between $\Phi_h^{hub}$ and  $\Phi_r$ instead of  calculating pairwise distances for samples in $\Phi_h^{hub}$ ?

---

> ### Author Response · Authors · 2023-11-19
> **Authors' Response**
>
> Thank you for your comments and we hope our responses address your concerns.
>
> > **Q1:** The authors indicate in Sec. 4.3 that the numbers of hub samples, i.e. $m_r$ and $m_g$ are far less than the number of samples of original sets, i.e. $n$. However, the specific conditions for the validity of this conclusion are not provided. From the experimental results in Table 2, the ratio of $O(m_r)$ or $O(m_g)$ to $O(n)$ is about 0.6, which is not consistent with the statement $m_r \ll n, m_g \ll n$.
>
> **Re:** We agree that the definition of "$\ll$" is subjective and may cause confusion. However, we hope to clarify that in our work, we used a threshold of $t=3$ (Sec. 5.4, "Choice of threshold $t$"), which results in a ratio of $O(m_r)$ or $O(m_g)$ to $O(n)$ of around 0.38 (Table 7 in the supplementary materials).
> We have removed the "$\ll$" notion and included example ratios in our revised paper.
>
> > **Q2a:** In Observation 4.1 and Figure 1, the authors roughly divide hubness values into three groups and claim that samples with similar hubness values are effective representative samples in P&R ratio calculation, which lacks generality and specific analysis. Further illustration is needed to explain why the hubness value split points are chosen as 12 and 24. Observation 4.2 and Figure 2 have the same issue.
>
> **Re:** First, we provide a further analysis to show that the performance of our eP&R is insensitive to the group split points (e.g., 12 and 24).
>
> - [Observation 4.1 and Fig. 1]. As shown in (a) and (b) below, the ratios of binary scores are similar for each hubness value, which validates the insensitivity of the choice of group split points.
>
> (a) All 70k images in the FFHQ dataset
> |Hubness Value| 1     | 2     | 3     | 4     | 5     | 6     | 7     | 8     | 9     | 10    | 11    | 12    | 13    | 14    | 15    | 16    | 17    | 18    | 19    | 20    | 21    | 22    | 23    | 24    |
> |-------|-------|-------|-------|-------|-------|-------|-------|-------|-------|-------|-------|-------|-------|-------|-------|-------|-------|-------|-------|-------|-------|-------|-------|-------|
> |Binary Score = 1| 9038  | 7489  | 5795  | 4579  | 3525  | 2741  | 2188  | 1842  | 1474  | 1244  | 931   | 818   | 679   | 593   | 508   | 425   | 361   | 305   | 272   | 246   | 200   | 177   | 153   | 160   |
> |All Samples| 13382 | 11292 | 8615  | 6815  | 5250  | 4158  | 3235  | 2664  | 2190  | 1791  | 1389  | 1231  | 1046  | 883   | 776   | 640   | 543   | 469   | 409   | 361   | 291   | 264   | 232   | 230   |
> |Ratio| 0.675 | 0.663 | 0.673 | 0.672 | 0.671 | 0.659 | 0.676 | 0.691 | 0.673 | 0.695 | 0.670 | 0.665 | 0.649 | 0.672 | 0.655 | 0.664 | 0.665 | 0.650 | 0.665 | 0.681 | 0.687 | 0.670 | 0.659 | 0.696 |
>
>
> (b) 70k images generated by StyleGAN2
> |Hubness Value| 1     | 2     | 3     | 4     | 5     | 6     | 7     | 8     | 9     | 10    | 11    | 12    | 13    | 14    | 15    | 16    | 17    | 18    | 19    | 20    | 21    | 22    | 23    | 24    |
> |-------|-------|-------|-------|-------|-------|-------|-------|-------|-------|-------|-------|-------|-------|-------|-------|-------|-------|-------|-------|-------|-------|-------|-------|-------|
> |Binary Score = 1| 10030 | 9150  | 7375  | 5858  | 4555  | 3664  | 2952  | 2266  | 1848  | 1515  | 1358  | 1120  | 911   | 800   | 646   | 522   | 456   | 374   | 350   | 306   | 261   | 228   | 187   | 178   |
> |All Samples| 12083 | 10988 | 8833  | 7015  | 5452  | 4402  | 3561  | 2743  | 2237  | 1835  | 1624  | 1344  | 1117  | 992   | 762   | 635   | 539   | 467   | 420   | 379   | 317   | 277   | 217   | 208   |
> |Ratio| 0.830 | 0.833 | 0.835 | 0.835 | 0.835 | 0.832 | 0.829 | 0.826 | 0.826 | 0.826 | 0.836 | 0.833 | 0.816 | 0.806 | 0.848 | 0.822 | 0.846 | 0.801 | 0.833 | 0.807 | 0.823 | 0.823 | 0.862 | 0.856 |
>
> - [Observation 4.2 and Fig. 2] As shown in (a) and (b) below, the ratios of hubness samples increase quickly to 1 with the increase of $|\phi'|$, which validates the insensitivity of the choice of group split points.
>
> (a) All 70k images in the FFHQ dataset
> |$\|\phi'\|$| 0     | 1     | 2     | 3     | 4     | 5     | >=6   |
> |-------|-------|-------|-------|-------|-------|-------|-------|
> |Hubness| 11922 | 8625  | 6290  | 4540  | 3516  | 2741  | 10818 |
> |Non-hubness| 13254 | 9089  | 6519  | 4598  | 3537  | 2749  | 10818 |
> |Ratio| 0.900 | 0.949 | 0.965 | 0.987 | 0.994 | 0.997 | 1     |
>
> (b) 70k images generated by StyleGAN2
> |$\|\phi'\|$| 0     | 1     | 2     | 3     | 4     | >=5   |
> |-------|-------|-------|-------|-------|-------|-------|
> |Hubness| 11792 | 6596  | 4136  | 2745  | 1860  | 5531  |
> |Non-hubness| 13482 | 7210  | 4328  | 2788  | 1874  | 5531  |
> |Ratio| 0.875 | 0.915 | 0.956 | 0.985 | 0.992 | 1.000 |
>
> We have included these Tables and relevant discussions in Table 10 and Table 11 in Sec. A.4 (appendix) of the revised paper.

---

> ### Author Response · Authors · 2023-11-19
> **Authors' Response 2**
>
> > **Q2b:** Whether this observation holds in many other datasets.
>
> **Re:** To demonstrate the generality of these observations, we include the corresponding results on the LSUN-Church dataset below, where our observations still hold.
>
> - [Observation 4.1]:
>
> (a) All 120k images in the LSUN-Church dataset
> |Hubness Value| 1     | 2     | 3     | 4     | 5     | 6     | 7     | 8     | 9     |
> |-------|-------|-------|-------|-------|-------|-------|-------|-------|-------|
> |Binary Score = 1| 51045 | 17414 | 6653  | 2906  | 1413  | 658   | 355   | 202   | 108   |
> |All Samples| 75650 | 25836 | 9866  | 4296  | 2074  | 1006  | 562   | 291   | 153   |
> |Ratio| 0.675 | 0.674 | 0.674 | 0.676 | 0.681 | 0.654 | 0.631 | 0.694 | 0.708 |
>
> (b) 100k images generated by StyleGAN2
> |Hubness Value| 1     | 2     | 3     | 4     | 5     | 6     | 7     | 8     | 9     | 10    |
> |-------|-------|-------|-------|-------|-------|-------|-------|-------|-------|-------|
> |Binary Score = 1| 24459 | 4658  | 1081  | 405   | 161   | 71    | 35    | 19    | 9     | 6     |
> |All Samples| 55401 | 10564 | 2385  | 957   | 371   | 154   | 81    | 38    | 21    | 12    |
> |Ratio| 0.441 | 0.441 | 0.453 | 0.423 | 0.434 | 0.461 | 0.432 | 0.500 | 0.429 | 0.500 |
>
> - [Observation 4.2]:
>
> (a) All 120k images in the LSUN-Church dataset
> |$\|\phi'\|$| 0     | 1     | 2     | 3     | 4     | 5     | >=6   |
> |-------|-------|-------|-------|-------|-------|-------|-------|
> |Hubness| 29986 | 18482 | 12819 | 9106  | 6749  | 5037  | 20441 |
> |Non-hubness| 32669 | 19615 | 13346 | 9213  | 6773  | 5040  | 20441 |
> |Ratio| 0.918 | 0.942 | 0.961 | 0.988 | 0.996 | 0.999 | 1.000 |
>
> (b) 100k images generated by StyleGAN2
> |$\|\phi'\|$| 0     | 1     | 2     | 3     | 4     | 5     | 6     | >=7   |
> |-------|-------|-------|-------|-------|-------|-------|-------|-------|
> |Hubness| 15731 | 7063  | 3984  | 2405  | 1505  | 963   | 728   | 2007  |
> |Non-hubness| 17590 | 7821  | 4158  | 2443  | 1513  | 965   | 729   | 2007  |
> |Ratio| 0.894 | 0.903 | 0.958 | 0.984 | 0.995 | 0.998 | 0.999 | 1.000 |
>
> We have included these Tables and relevant discussions in Table 12 and Table 13 in Sec. A.4 (appendix) of the revised paper.
>
> > **Q3:** In Sec. 4.2, the authors point out the insensitivity of hubness-aware sampling to exact k-nearest neighbor (k-NN) results, which might be confusing since in Table 4, the Precision and Recall change greatly when k is taken from 3 to 10. Therefore, specific mathematical descriptions are required to substantiate this viewpoint.
>
> **Re:** We believe there is a misunderstanding. In Sec. 4.2, by "insensitivity", we meant that our eP&R metric is insensitive to the choice of *exact* vs. *approximate* k-NN algorithms but not the choice of $k$. Thus, it does not contradict the results in Table 4. We have clarified this in Sec. 4.2 of the revised paper.
>
> In our paper, we use HNSW [1] as the approximate k-NN algorithm, which achieves high accuracies above 99% and has been widely used by the community. Please see [1] for more details about the validity of the algorithm.
>
> [1] Yu A Malkov and Dmitry A Yashunin. Efficient and robust approximate nearest neighbor search using hierarchical navigable small world graphs. IEEE transactions on pattern analysis and machine intelligence, 42(4):824–836, 2018.
>
> > **Q4:** The font size of the annotations in Figure 1 and Figure 2 is too small to identify clearly. Besides, the explanation for (a) in Figure 2 is unclear, which can be directly replaced by 'hubness' and 'non-hubness'.
>
> **Re:** We have fixed these in the revised paper.

---

> ### Author Response · Authors · 2023-11-19
> **Authors' Response 3**
>
> > **Q5:** In Sec. 4.3 in the third stage of complexity analysis for eP&R, why calculating pairwise distances for samples between $\mathbf{\Phi}_r^{hub}$ and $\mathbf{\Phi}_r$ instead of calculating pairwise distances for samples in $\mathbf{\Phi}_r^{hub}$?
>
> **Re:** Good question! As the table below shows, we calculate the pairwise distances between $\mathbf{\Phi}_r^{hub}$ and $\mathbf{\Phi}_r$ as it provides lower approximation errors than calculating pairwise distances for samples in $\mathbf{\Phi}_r^{hub}$.
> We conjecture the reason is that $\mathbf{\Phi}_r^{hub}$ is much sparser than $\mathbf{\Phi}_r$ and thus the pairwise distances for samples in it will be much larger than those of the original P&R, resulting in much larger $k$-NN hyperspheres that increase the approximation error.
> The same conclusion holds for $\mathbf{\Phi}_g^{hub}$ and $\mathbf{\Phi}_g$.
>
> We have included the table and relevant discussions in Table 14, Sec. A.5 (appendix) of the revised paper.
>
> |       | Pairwise distances between $\mathbf{\Phi}\_r^{hub}$ and $\mathbf{\Phi}\_r^{hub}$|       |               |                | Pairwise distances between $\mathbf{\Phi}\_r^{hub}$ and $\mathbf{\Phi}\_r$   | | |                 |
> | ----- | -------------------------------------------------  | ----- | --------------| ---------------|  ---------------------------------  | ------| --------------| --------------|
> | t     | P                                                  | R     | P Error(\%) | R Error(\%)  | P                                   | R     | P Error(\%) | R Error(\%) |
> | 1     | 0.713                                              | 0.484 | 0.4           | 1.8            | 0.717                               | 0.500 | 0.3           | 0.0           |
> | 2     | 0.723                                              | 0.506 | 1.0           | 2.6            | 0.718                               | 0.501 | 0.3           | 0.0           |
> | 3     | 0.746                                              | 0.534 | 4.2           | 8.3            | 0.719                               | 0.501 | 0.4           | 0.2           |
> | 4     | 0.768                                              | 0.562 | 7.3           | 14.0           | 0.726                               | 0.507 | 1.6           | 0.6           |
> | 5     | 0.787                                              | 0.588 | 9.9           | 19.3           | 0.730                               | 0.515 | 1.9           | 0.6           |

---

### Meta-Review · Area_Chair_iCBs · 2023-12-05

**Metareview:**

- Claims and findings:
This submission introduces an efficient way of compute Precision and Recall metrics to evaluate generative models. The proposed efficient PR metrics are almost identical to original PR metric while being more efficient to compute. The submission achieves this by proposed the use of a hubness-aware sampling method that avoids redundancies in computation. Experiments confirm the effectiveness of the approach.

- Strengths:

Reviewers have highlighted that this work proposes efficient precision and recall (eP&R) metrics for assessing generative models to approximate results as the original P&R metrics with lower consumption in time and space. In addition, reviewers have also mentioned that the ablation study and the experiment on robustness against the truncation trick are a nice addition to the experiments section. The experiments convincingly demonstrate savings on compute time and storage for real world datasets, across a variety of model architectures.


- Weaknesses:

Reviewers have pointed out that the proposed speedup is specific to one particular way of P&R estimation i.e. using the Kynkaanniemi et al 2019 method based on nearest neighbors. This method only gives two scalar values corresponding to P and R. In contrast, Simon et al. ICML 2019 method gives the whole PR curve. In addition, it's also unclear how the proposed method would behave in the 0 and 1 corner cases (authors have referred to updated revision which seems to not exist).


- Missing in submission:

I believe this submission could benefit from a larger set of experiments on bigger datasets where the efficiency claims really shine. In addition, it seems that authors have not updated either the submission or the appendix following reviewers suggestions.

**Justification For Why Not Higher Score:**

Although authors have provided a rebuttal the refer to updated appendix and revisions that do not seem to exists.

**Justification For Why Not Lower Score:**

The idea of making more efficient metrics to evaluate generative models is important and this paper deserved credit for it.

---

### Decision · Program_Chairs · 2024-01-16

Reject